# Neural Field Discovery Disentangles Equivariance in Interacting Dynamical Systems

## Abstract

Systems of interacting objects often evolve under the influence of underlying field effects that govern their dynamics, *e.g.* electromagnetic fields in physics, or map topologies and traffic rules in traffic scenes. While the interactions between objects depend on local information, the underlying fields depend on global states. Pedestrians and vehicles in traffic scenes, for example, follow different traffic rules and social norms depending on their absolute geolocation. The entanglement of global and local effects makes recently popularized equivariant networks inapplicable, since they fail to capture global information. To address this, in this work, we propose to *disentangle* local object interactions –which are equivariant to global roto-translations and depend on relative positions and orientations– from external global field effects –which depend on absolute positions and orientations. We theorize the presence of latent fields, which we aim to discover *without* directly observing them, but infer them instead from the dynamics alone. We propose neural fields to learn the latent fields, and model the interactions with equivariant graph networks operating in local coordinate frames. We combine the two components in a graph network that transforms field effects in local frames and operates solely there. Our experiments show that we can accurately discover the underlying fields in charged particles settings, traffic scenes, and gravitational n-body problems, and effectively use them to learn the system and forecast future trajectories.

## 1 Introduction

Systems of interacting objects are omnipresent in nature, with examples ranging from the sub-atomic to the astronomical scale –including colliding particles and n-body systems of celestial objects– as well as human-centric settings like traffic scenes, governed by social dynamics. The majority of these systems does not evolve in a vacuum, they instead evolve under the influences of underlying fields. For example, electromagnetic fields may govern the dynamics of charged particles. In traffic scenes, the road network and traffic rules govern the actions of traffic scene

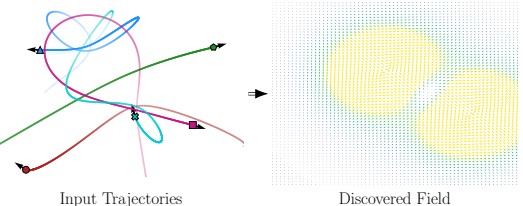

Input Trajectories          Discovered Field

Figure 1: N-body system simulation with underlying gravitational field. We uncover fields that underlie interacting systems using trajectories only.

participants. N-body systems might swirl around supermassive black holes that create gravitational fields. Earlier work on learning interacting systems proposed graph networks (Kipf et al., 2018; Battaglia et al., 2016; Sanchez-Gonzalez et al., 2020), while state-of-the-art methods for interacting systems propose *equivariant* graph networks (Walters et al., 2021; Satorras et al., 2021; Kofinas et al., 2021; Brandstetter et al., 2022) to model dynamics while respecting their underlying symmetries. These networks exhibit increased robustness and performance, while maintaining parameter efficiency due to weight sharing. They are, however, not compatible with underlying field effects, since they can only capture local states, such as relative positions, while fields depend on absolute states (positions or orientations). In other words, *global fields violate the strict equivariance hypothesis*. We are, thus, in need of an augmented notion of equivariance that encapsulates both local and global effects.

In many real-world settings, strict $SE(3)$ equivariance, –equivariance to the special Euclidean group of translations and rotations– does not hold. A function $f$ that predicts trajectories in interacting systems

is equivariant to translations if $f(\mathbf{x}) + \boldsymbol{\tau} = f(\mathbf{x} + \boldsymbol{\tau})$ for a translation vector $\boldsymbol{\tau}$, and equivariant to rotations if $\mathbf{Q}f(\mathbf{x}) = f(\mathbf{Q}\mathbf{x})$ for a rotation matrix $\mathbf{Q}$. That is, even if the symmetries exist in a particular setting, they only manifest themselves in local interactions, yet they are entangled with global effects that stem from absolute states. For instance, pedestrians and vehicles in traffic scenes operate within egocentric perspectives; individual local coordinate frames allow for differentiating objects based on relative states. Objects, however, behave differently depending on their absolute geolocation, *e.g.* people in different countries or cities follow different traffic rules and social norms, and exhibit different driving habits. In such cases, global transformations are not even properly defined; it would be illogical to perform a translation or a rotation within the global coordinate frame of the Earth, since this would coincide with a different location on the globe.

Even within a local region of space, though, certain effects may only depend on global object states. N-body systems from physics, for example, exhibit roto-translational symmetries, since gravitational forces only depend on relative positions. Dynamics, however, may be influenced by external gravitational fields, which are either unknown or not subject to transformations. Thus, strict equivariance is violated, since equivariant object interactions are *entangled* with global field effects.

In this work, we *disentangle* the interactions between objects –which are equivariant to global roto-translations and depend on relative positions and orientations– from external field effects –which depend on absolute positions and orientations. Furthermore, we propose neural fields to model the underlying field effects. Neural fields depend on absolute positions, and potentially orientations, and predict latent *force fields*. We model discrete object interactions with equivariant local coordinate frame graph networks, and continuous field effects with neural fields.

We make the following contributions. First, we introduce the notion of *entangled equivariance* that intertwines global and local effects, and propose a novel architecture that disentangles equivariant local object interactions from global field effects. Second, we introduce neural fields to discover global latent fields in interacting dynamical systems, and infer them by observing the dynamics alone. Third, we propose an approximately equivariant graph network that extends local coordinate frame graph networks by introducing an auxiliary origin node, resulting in a mixture of global and local coordinate frames. Finally, we conduct experiments on a number of field settings, and observe that explicitly modelling fields is mandatory for effective future forecasting, while their unsupervised discovery opens a window for model explainability.

We term our method *Aether*, inspired by the postulated medium that permeates all throughout space and allows for the propagation of light.

## 2 BACKGROUND

In this section, we introduce background knowledge on interacting systems, local coordinate frame graph networks, and neural fields, which will serve as a foundation for our method.

### 2.1 INTERACTING DYNAMICAL SYSTEMS

An interacting dynamical system comprises trajectories of $N$ objects, recorded for $T$ timesteps. The snapshot of the $i$-th object at timestep $t$ describes the state $\mathbf{x}_i^t = [\mathbf{p}_i^t, \mathbf{u}_i^t], i \in \{1, \ldots, N\}, t \in \{1, \ldots, T\}$, where $\mathbf{p}$ denotes the position and $\mathbf{u}$ denotes the velocity, using $[\cdot, \cdot]$ to denote vector concatenation along the feature dimension. Interacting dynamical systems can be naturally formalized as spatio-temporal geometric graphs (Battaglia et al., 2016; Kipf et al., 2018; Graber & Schwing, 2020), $\mathcal{G} = \{\mathcal{G}^t\}_{t=1}^T$, with graph snapshots $\mathcal{G}^t = (\mathcal{V}^t, \mathcal{E}^t)$ at different time steps. The set of graph nodes $\mathcal{V}^t = \{v_1^t, \ldots, v_N^t\}$ describes the objects in the system; $v_i^t$ corresponds to $\mathbf{x}_i^t$. The set of edges $\mathcal{E}^t \subseteq \{(v_j^t, v_i^t) \mid (v_j^t, v_i^t) \in \mathcal{V}^t \times \mathcal{V}^t\}$ describes pair-wise object interactions; $(v_j^t, v_i^t)$ corresponds to an interaction from node $j$ to node $i$. Finally, we denote the graph neighbors of node $v_i$ with $\mathcal{N}(i)$.

### 2.2 LOCAL COORDINATE FRAME GRAPH NETWORKS

Local coordinate frame graph networks have been popularized in recent years (Kofinas et al., 2021; Luo et al., 2022) as a method to achieve SE(3) equivariance, due to their low computational overhead and high performance. Kofinas et al. (2021) proposed LoCS and introduced local coordinate frames

for all node-objects at all timesteps. They define augmented node states $\mathbf{v}_i^t = [\mathbf{p}_i^t, \boldsymbol{\omega}_i^t, \mathbf{u}_i^t]$, where $\boldsymbol{\omega}_i^t$ denotes the angular position of node $i$ at timestep $t$. Each local coordinate frame is translated to match the target object's position and rotated to match its orientation. Considering the representation of node $j$ in the local coordinate frame of node $i$, they first compute the relative positions $\mathbf{r}_{j,i}^t = \mathbf{p}_j^t - \mathbf{p}_i^t$ and then they rotate the state using the matrix representation of the angular position $\mathbf{Q}(\boldsymbol{\omega}_i^t)$:

$$\mathbf{v}_{j|i}^t = \tilde{\mathbf{R}}(\boldsymbol{\omega}_i^t)^\top [\mathbf{r}_{j,i}^t, \boldsymbol{\omega}_j^t, \mathbf{u}_j^t], \tag{1}$$

where $\tilde{\mathbf{R}}(\boldsymbol{\omega}_i^t) = \mathbf{Q}(\boldsymbol{\omega}_i^t) \oplus \mathbf{Q}(\boldsymbol{\omega}_i^t) \oplus \mathbf{Q}(\boldsymbol{\omega}_i^t)$, and $\oplus$ denotes a direct sum. LoCS then proposes a graph neural network (Scarselli et al., 2008; Li et al., 2016; Gilmer et al., 2017) that uses local states:

$$\mathbf{h}_{j,i}^t = f_e\left(\left[\mathbf{v}_{j|i}^t, \mathbf{v}_{i|i}^t\right]\right), \tag{2}$$

$$\boldsymbol{\Delta}\mathbf{x}_{i|i}^{t+1} = f_v\left(g_v\left(\mathbf{v}_{i|i}^t\right) + \frac{1}{|\mathcal{N}(i)|}\sum_{j \in \mathcal{N}(i)}\mathbf{h}_{j,i}^t\right), \tag{3}$$

where $f_v$, $f_e$, and $g_v$ are MLPs. The output of this graph network comprises differences in positions and velocities from the previous time step, in the local frame of each object. Since these outputs are invariant, LoCS performs an inverse transformation to convert them back to the global coordinate frame and achieve equivariance, $\mathbf{x}_i^{t+1} = \mathbf{x}_i^t + \mathbf{R}(\boldsymbol{\omega}_i^t) \cdot \boldsymbol{\Delta}\mathbf{x}_{i|i}^{t+1}$, where $\mathbf{R}(\boldsymbol{\omega}_i^t) = \mathbf{Q}(\boldsymbol{\omega}_i^t) \oplus \mathbf{Q}(\boldsymbol{\omega}_i^t)$.

### 2.3 NEURAL FIELDS

Finally, we make a brief introduction to neural fields. Neural fields, or coordinate-based MLPs, are a class of neural networks that parameterize fields using neural networks (Xie et al., 2022). They take as input spatial or temporal coordinates and predict some quantity. Neural fields can learn prior behaviors and generalize to new fields via conditioning on a latent variable $\mathbf{z}$ that encodes the properties of a specific field. Perez et al. (2018) proposed Feature-wise Linear Modulation (FiLM), a conditioning mechanism that modulates a signal. It comprises two sub-networks $\alpha, \beta$ that perform multiplicative and additive modulation to the input signal, and can be described by FiLM$(\mathbf{h}, \mathbf{z}) = \alpha(\mathbf{z}) \odot \mathbf{h} + \beta(\mathbf{z})$, where $\mathbf{z}$ is the conditioning latent variable, $\mathbf{h}$ is the signal to be modulated, and $\alpha, \beta$ are MLPs that scale the signal, and add a bias term, respectively.

## 3 METHOD

In this section, we present our method, termed *Aether*. First, we describe the notion of entangled equivariance, and introduce our architecture that disentangles global field effects from local object interactions. Then, we continue with the description of the neural field that infers latent fields by observing the dynamics alone. Finally, we formulate approximately equivariant global-local coordinate frame graph networks. We note that throughout this work, we focus on 2-dimensional settings, and fields that are unaffected by the observable objects and their interactions thereof.

### 3.1 AETHER

Interacting dynamical systems rarely evolve in a vacuum, rather they evolve under the influence of external field effects. While object interactions depend on local information, the underlying fields depend on global states. On the one hand, locality in object interactions stems from the fact that dynamics obey a number of symmetries. By extension, object interactions are equivariant to a particular group of transformations. On the other hand, field effects are non-local; they depend on absolute object states. Thus, strict equivariance is violated, since equivariant object interactions are entangled with global field effects. We refer to this phenomenon as *entangled equivariance*.

As an example, in fig. 2 we observe a system of 2 objects that evolve under the influence of a gravitational field. The

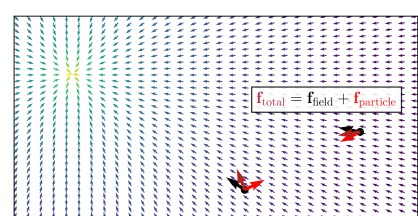

Figure 2: Two objects in a gravitational field. We only observe the total force exerted at each particle, *i.e.* the sum of equivariant particle forces and global field effects.

arrows positioned *on* the objects represent the forces exerted on them. One constituent of the net force is caused by the object interactions, and is thus equivariant, while the other can be attributed to the gravitational pull. However, we can only observe the net force exerted at each particle, *i.e.* the sum of equivariant particle forces and non-equivariant force field effects. Hence, in this system, we say that *equivariance is entangled*.

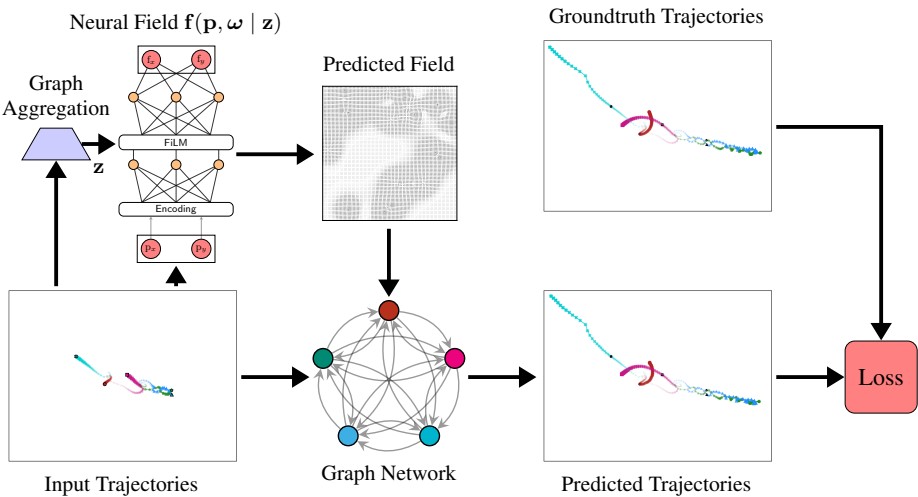

Figure 3: The pipeline of our method, *Aether*.

We now propose our architecture that disentangles local object interactions from global field effects. We model the object interactions with local coordinate frame graph networks (Kofinas et al., 2021), and field effects with neural fields. During training, and given a multitude of input systems, neural fields will, in principle, be able to isolate global from local effects, since only global effects are recurring phenomena. We hypothesize that field effects can be attributed to force fields, and therefore, our neural fields learn to discover *latent force fields*. The pipeline of our method is shown in fig. 3. Our inputs comprise states $\mathbf{x}_i^t$ for trajectories of $N$ objects for $T$ timesteps. Since neural fields model global field effects, they depend on absolute states. Thus, we feed positions $\mathbf{p}_i^t$, and potentially orientations $\boldsymbol{\omega}_i^t$, of the trajectories as input to a neural field that predicts latent forces $\mathbf{f}_i^t = \mathbf{f}(\mathbf{p}_i^t, \boldsymbol{\omega}_i^t)$.

The predicted field forces can be now considered part of the node states, and further, they can be treated similarly to other state variables like velocities; as vectors, forces are translation invariant and rotation equivariant. Thus, moving onward, we can treat the problem setup as if we were once again back in the strict equivariance regime. We append the predicted forces $\mathbf{f}_i^t$ for each node-object $i$ and each timestep $t$ to the node states, and transform them to corresponding local coordinate frames, similarly to eq. (1). Namely, the force exerted on node $j$, expressed in the local coordinate frame of node $i$ is computed as:

$$\mathbf{f}_{j|i}^t = \mathbf{Q}^\top(\boldsymbol{\omega}_i^t)\mathbf{f}_j^t. \tag{4}$$

We feed the new local node states to a local coordinate frame graph network as follows:

$$\mathbf{h}_{j,i}^t = f_e\left(\left[\mathbf{v}_{j|i}^t, \boxed{\mathbf{f}_{j|i}^t}, \mathbf{v}_{i|i}^t, \boxed{\mathbf{f}_{i|i}^t}\right]\right) \tag{5}$$

$$\boldsymbol{\Delta}\mathbf{x}_{i|i}^{t+1} = f_v\left(g_v\left(\left[\mathbf{v}_{i|i}^t, \boxed{\mathbf{f}_{i|i}^t}\right]\right) + \frac{1}{|\mathcal{N}(i)|}\sum_{j\in\mathcal{N}(i)}\mathbf{h}_{j,i}^t\right) \tag{6}$$

$$\mathbf{x}_i^{t+1} = \mathbf{x}_i^t + \mathbf{R}(\boldsymbol{\omega}_i^t)\cdot\boldsymbol{\Delta}\mathbf{x}_{i|i}^{t+1}, \tag{7}$$

where $\mathbf{R}(\boldsymbol{\omega}_i^t) = \mathbf{Q}(\boldsymbol{\omega}_i^t)\oplus\mathbf{Q}(\boldsymbol{\omega}_i^t)$. The equations above are similar to eqs. (2) and (3), with the addition of the highlighted parts that denote the predicted forces expressed at local coordinate frames. In practice, we closely follow (Kipf et al., 2018; Graber & Schwing, 2020; Kofinas et al., 2021) and formulate our model as a variational autoencoder (Kingma & Welling, 2014; Rezende et al., 2014) with latent edge types. The exact details are presented in appendix A.2.2.

## 3.2 Field discovery

In many settings, fields might not be directly observable for us to probe them at will and use them for supervision. For example, astronomical observations of solar systems and galaxies might not include black holes, yet we can observe their effects. Moreover, fields are often not even measurable or quantifiable, or they are not explicitly defined. For instance, *social "fields"* that guide traffic, cannot be measured or defined explicitly, but we can safely assume they exist. Motivated by these observations, we design an architecture that performs *unsupervised field discovery*, while solving the surrogate supervised task of trajectory forecasting.

In this work, we aim to discover two different types of fields, which we term "static" and "dynamic" fields. Static fields refer to settings in which we have a single field shared throughout the whole dataset. On the other hand, dynamic fields refer to settings in which we have a different field for each input system, and fields also differ between train, validation, and test sets.

We now describe neural fields, used in this work, to model the underlying field effects. Neural fields depend on absolute positions, and potentially orientations, and predict latent force fields. When dealing with *static* fields, we use an *unconditional* neural field, *i.e.* a neural field that is a function only of the input positions-orientations, as the field values are common across data samples. In contrast, for *dynamic* fields, we use a *conditional* neural field, *i.e.* a neural field that also depends on a latent vector $\mathbf{z} \in \mathbb{R}^{D_z}$ that represents the underlying field. The latent $\mathbf{z}$ will be inferred from the input trajectories and can be thought of as representing unusual non-equivariant dynamics. We use $\mathbf{z}$ to explicitly condition the neural field, and thus, its general form is $\mathbf{f} : \mathbb{R}^2 \times \mathcal{S}^1 \times \mathbb{R}^{D_z} \rightarrow \mathbb{R}^2$.

**Static fields** We start with the description of unconditional neural fields used in static field settings, since conditional neural fields share the same backbone. First, we encode the input positions using Gaussian random Fourier features (Gaussian RFF) (Tancik et al., 2020). They are defined as:

$$\gamma(\mathbf{p}) = [\cos(2\pi\mathbf{B}\mathbf{p}), \sin(2\pi\mathbf{B}\mathbf{p})]^\top, \tag{8}$$

where $\mathbf{p} \in \mathbb{R}^{D_{\text{in}}}$ are the input coordinates, and $\mathbf{B} \in \mathbb{R}^{D_c \times D_{\text{in}}}$ is a matrix with entries sampled from a Gaussian distribution, $\mathbf{B}_{kl} \sim \mathcal{N}(0, \sigma^2)$. The variance $\sigma^2$ can be chosen per task with a hyperparameter sweep.

We use a unit vector representation for the orientations $\theta$, $\boldsymbol{\theta} = [\cos\theta, \sin\theta]^\top$. Following Kofinas et al. (2021), we use the angles of the velocity vectors as a proxy for the orientations. Then, we use a linear layer to encode the orientation vectors, $\delta(\boldsymbol{\theta}) = \mathbf{W}_\omega \boldsymbol{\theta}$. We finally concatenate the encoded positions and orientations in a single vector that is being fed as input to the neural field.

After encoding the input coordinates, we use a 3-layer MLP with SiLU (Ramachandran et al., 2018) activations in-between, that outputs a latent force field, $\mathbf{f}(\mathbf{p}, \boldsymbol{\theta}) = \text{MLP}([\gamma(\mathbf{p}), \delta(\boldsymbol{\theta})])$.

**Dynamic fields** The neural fields used to model the dynamic fields are conditioned on a latent vector representation $\mathbf{z} \in \mathbb{R}^{D_z}$ that describes prior knowledge about the underlying field, and are defined as $\mathbf{f}(\mathbf{p}, \boldsymbol{\theta} \mid \mathbf{z})$. In our case, the latent representation should "summarize" the input graph such that it isolates only global effects from the field. To that end, we employ a simple global spatio-temporal attention mechanism, similar to Li et al. (2016), that aggregates the input system in a latent vector representation. First, we define object embeddings $\mathbf{o}_i = \text{GRU}(\mathbf{W}_g \mathbf{x}_i^{1:T})$, where $\mathbf{W}_g$ is a matrix used to linearly transform the inputs, and GRU is the Gated Recurrent Unit (Cho et al., 2014). We also define temporal embeddings $\mathbf{t} = \text{PE}(t)$, where PE are positional encodings (Vaswani et al., 2017). The aggregation is then defined as follows:

$$\mathbf{z} = \sum_{i,t} \text{softmax}\left(f_a\left(\mathbf{s}_i^t\right)\right) \cdot f_b\left(\mathbf{s}_i^t\right), \quad \text{with} \quad \mathbf{s}_i^t = \left[\mathbf{x}_i^t, \mathbf{o}_i\right] + \mathbf{t}, \tag{9}$$

where $f_a : \mathbb{R}^{D_s} \rightarrow \mathbb{R}, f_b : \mathbb{R}^{D_s} \rightarrow \mathbb{R}^{D_z}$ are 2-layer MLPs with SiLU activations Ramachandran et al. (2018) in between.

After having obtained a latent vector representation $\mathbf{z}$ that summarizes the input system, we condition the neural field using FiLM (Perez et al., 2018). We include FiLM layers after the first two linear layers of the neural field. The exact details are presented in appendix A.2.1.

### 3.3 APPROXIMATE EQUIVARIANCE WITH GLOBAL-LOCAL COORDINATE FRAMES

Equivariant neural networks cannot capture non-local information, such as global field effects. In this work, we explicitly aim to discover these fields and disentangle them from local object interactions. An alternative, or rather complementary approach, would be to directly combine global and local information, following the recently proposed notion of *approximate equivariance* (Wang et al., 2022). Starting from LoCS (Kofinas et al., 2021), we can combine global information and still operate in local coordinate frames by defining an auxiliary node-object corresponding to the global coordinate frame, *i.e.* an object positioned at the origin, and oriented to match the x-axis.

Similar to all objects in the system, the full state of the origin node $\mathcal{O}$ comprises the concatenation of its position and velocity, $\mathbf{x}_{\mathcal{O}} = [\mathbf{p}_{\mathcal{O}}, \mathbf{u}_{\mathcal{O}}] = (0, 0, 1, 0)^{\top}$. We temporarily add this node in the geometric graph, and we further connect it with edges *to* all other objects. Thus, the new vertex set is $\mathcal{V}' = \mathcal{V} \cup \{\mathcal{O}\}$, and the new edge set is $\mathcal{E}' = \mathcal{E} \cup \{e_{\mathcal{O},i} \forall i \in \mathcal{V}\}$. The origin state can be expressed in local coordinate frames using similar transformations to eq. (1). The origin transformed in the local coordinate frame of the $i$-th object is computed as follows:

$$\mathbf{v}_{\mathcal{O}|i}^t = \mathbf{R}_i^{t\top} \left[ \mathbf{p}_{\mathcal{O}}^t - \mathbf{p}_i^t, \mathbf{u}_{\mathcal{O}}^t \right] = \mathbf{R}_i^{t\top} \left[ -\mathbf{p}_i^t, \mathbf{u}_{\mathcal{O}}^t \right]. \tag{10}$$

Since graph networks are permutation equivariant, we have to distinguish between this auxiliary origin node and the other nodes, otherwise, the origin would be treated as an actual object, and there is no way for the network to understand which the origin is. Hence, we augment each object's state with the origin node information expressed in local coordinate frames, extending eqs. (2) and (3) to

$$\mathbf{h}_{j,i}^t = f_e \left( \left[ \mathbf{v}_{j|i}^t, \mathbf{v}_{i|i}^t, \boxed{\mathbf{v}_{\mathcal{O}|i}^t} \right] \right), \tag{11}$$

$$\mathbf{\Delta x}_{i|i}^{t+1} = f_v \left( g_v \left( \left[ \mathbf{v}_{i|i}^t, \boxed{\mathbf{v}_{\mathcal{O}|i}^t} \right] \right) + \frac{1}{|\mathcal{N}(i)|} \sum_{j \in \mathcal{N}(i)} \mathbf{h}_{j,i}^t \right). \tag{12}$$

This approach allows us to remove the origin node from the actual graph, and push the information in the node states. We term this method *G-LoCS* (**G**lobal-**Lo**cal **C**oordinate Frame**S**). In practice, similar to Aether, we formulate our model as a variational autoencoder (Kingma & Welling, 2014; Rezende et al., 2014) with latent edge types. The full details are presented in appendix A.2.3. Finally, in practice, we integrate G-LoCS in Aether, since it can enhance the performance of our method.

## 4 RELATED WORK

**Equivariant graph networks** The seminal works of (Cohen & Welling, 2016; 2017; Worrall et al., 2017) introduced equivariant convolutional neural networks and demonstrated effectiveness, robustness, and increased parameter efficiency. Recently, many works have proposed equivariant graph networks (Schütt et al., 2017; Thomas et al., 2018; Fuchs et al., 2020; Walters et al., 2021; Satorras et al., 2021; Kofinas et al., 2021; Brandstetter et al., 2022; Luo et al., 2022). Walters et al. (2021) propose rotationally equivariant continuous convolutions for trajectory prediction. Satorras et al. (2021) propose a computationally efficient equivariant graph network that leverages invariant euclidean distances between node pairs. Kofinas et al. (2021) introduce roto-translated local coordinate frames for all objects in an interacting system and propose equivariant local coordinate frame graph networks. Brandstetter et al. (2022) generalize equivariant graph networks using steerable MLPs (Thomas et al., 2018) and incorporate geometric and physical information in message passing. Equivariant graph networks differ from our work since they cannot capture non-local information, while our work disentangles equivariant local interactions from global effects and captures them both.

**Approximate equivariance** Recently, a number of works has proposed to shift away from strict equivariance, in what (Wang et al., 2022) termed as *approximate equivariance*. Wang et al. (2022) propose approximately equivariant networks for dynamical systems, by relaxing equivariance constraints in group convolutions and steerable convolutions. van der Ouderaa et al. (2022) propose to relax strict equivariance by interpolating between equivariant and non-equivariant operations, using non-stationary kernels that also depend on the absolute input group element. Romero & Lohit (2021) propose Partial G-CNNs that learn layer-wise partial equivariances from data. We note that

even though these works and approximate equivariance share similarities with our work, our notion of disentangled equivariance is conceptually different. That is because related work uses the term approximate equivariance to denote that equivariance is "broken" due to noise or imperfections, while our work disentangles the system dynamics that are actually equivariant, from the global field effects that are not, and in fact, might be unaffected by such transformations.

**Neural fields**   Neural fields have recently exploded in popularity in 3D computer vision, popularized by NeRF (Mildenhall et al., 2021). Since MLPs are universal function approximators (Hornik et al., 1989), neural fields parameterized by MLPs can, in principle, encode continuous signals at arbitrary resolution. However, neural networks can suffer from "spectral bias" (Rahaman et al., 2019; Basri et al., 2020), *i.e.* they are biased to fit functions with low spatial frequency. To address this issue, a number of solutions have been proposed. Tancik et al. (2020) leverage Neural Tangent Kernel (NTK) theory and propose Random Fourier Features (RFF), showing that they can overcome the spectral bias. They also show that RFF are a generalization of positional encodings, popularized in recent years in natural language processing by Transformers (Vaswani et al., 2017). Concurrently, Sitzmann et al. (2020) proposed SIREN, neural networks with sinusoidal activation functions. While neural fields have been used extensively in computer vision problems including 3D scene reconstruction (Park et al., 2019; Mescheder et al., 2019) and differentiable rendering (Sitzmann et al., 2019; Mildenhall et al., 2021), they have not seen wide usage in dynamical systems. Notably, Raissi et al. (2019) proposed Physics-Informed Neural Networks (PINNs), neural PDE solvers based on neural fields.

## 5   EXPERIMENTS

We evaluate our proposed method, *Aether*, on settings that include static as well as dynamic fields. First, we explore charged particles (Kipf et al., 2018) that evolve under the effect of a static electrostatic field. Then, we evaluate our method on a subset of inD (Bock et al., 2020) that contains a single location, and thus a static field as well. Finally, we explore gravitational n-body problems (Brandstetter et al., 2022) with dynamic fields.

We compare our method against dNRI (Graber & Schwing, 2020) and LoCS (Kofinas et al., 2021). DNRI (Graber & Schwing, 2020) is a graph network operating in global coordinates, and is, in principle, able to uncover both the global and the local dynamics. It is formulated as a VAE (Kingma & Welling, 2014; Rezende et al., 2014) with latent edge types and explicitly infers a latent graph structure. LoCS (Kofinas et al., 2021), on the other hand, operates in local coordinates, and is, thus, unable to uncover the global dynamics. For all methods, we use their publicly available source code. Our source code, data, and models will be shared upon publication.

Our architecture and experimental setup closely follow Graber & Schwing (2020); Kofinas et al. (2021). Unless specified differently, our neural field has a hidden size of 512. In charged particles and in n-body problems, we only use positions as input to the neural field, while in traffic scenes we also use orientations. The full implementation details are presented in appendix A.2.2. In all settings, we report the mean squared error of positions and velocities over time, $E(t) = \frac{1}{ND} \sum_{n=1}^{N} \|\mathbf{x}_n^t - \hat{\mathbf{x}}_n^t\|_2^2$.

Here we demonstrate indicative visualizations, and provide more extensive qualitative results in appendix C.

### 5.1   CHARGED PARTICLES

First, we study the effect of static fields, *i.e.* a single field across all train, validation, and test simulations. We extend the charged particles dataset from Kipf et al. (2018) by adding a number of immovable sources. Overall, these sources act like regular particles, exerting forces on the observable particles, except we ignore any forces exerted to them, and fix their positions and velocities to zero. We use $M = 20$ "source" particles and $N = 5$ "observable" particles. Sources particles can in principle have different charge magnitudes than the observable particles, but for simplicity we set all magnitudes to $|q| = 1$. All particles carry either positive or negative charges with equal probability. Following Satorras et al. (2021); Fuchs et al. (2020); Kofinas et al. (2021), we remove virtual borders that cause elastic collisions. We generate a dataset of 50,000 simulations for training, 10,000 for validation and 10,000 for testing. Following Kipf et al. (2018), each simulation lasts for 49 timesteps. During inference, we use the first 29 steps as input and predict the remaining 20 steps.

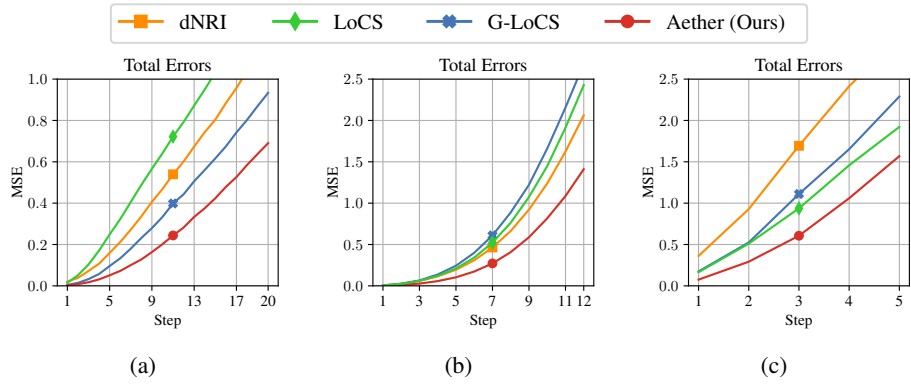

Figure 4: Results on (a) charged particles, (b) inD, and (c) gravity.

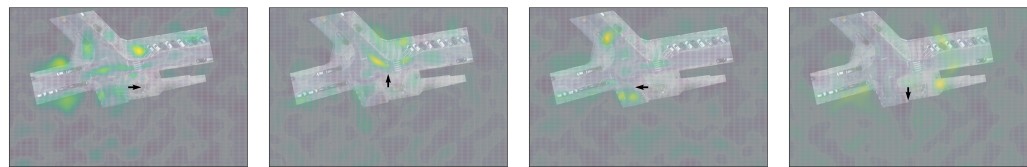

Figure 6: Discovered field on inD (Bock et al., 2020). For simplicity, we only visualize the field for discrete input orientations in $C_4 = \left\{0, \frac{\pi}{2}, \pi, \frac{3\pi}{2}\right\}$.

We compare our method against dNRI and LoCS, as well as G-LoCS, which should in principle be able to model both local and global dynamics effectively. We plot MSE in fig. 4a and $L_2$ errors in fig. 20, and visualize the learned field in fig. 5. We visualize sample predictions in fig. 8a, and showcase more predicted trajectories in appendix C.1. We observe that equivariant methods like LoCS perform poorly, while the approximately equivariant G-LoCS performs much better than equivariant and non-equivariant methods. Aether outperforms all other methods, demonstrating that it can *disentangle equivariance*. Furthermore, as shown in fig. 5, and fig. 12 in appendix C.1.1, *Aether can effectively discover the underlying field*.

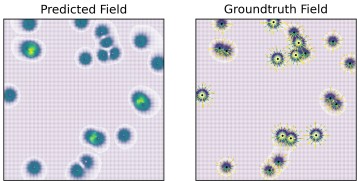

Figure 5: Learned Field (left) in charged particle settings compared to groundtruth (right).

## 5.2 TRAFFIC SCENES

Next, we study the effectiveness of static field discovery in traffic scenes. We use inD (Bock et al., 2020), a dataset with real-world traffic scenes that comprises trajectories of pedestrians, vehicles, and cyclists, recorded at 4 different locations. We create a subset that contains scenes from a single location. Namely, we choose "Frankenburg, Aachen", since it is the location with most interactions in the dataset. The subset corresponds to 12 recordings; we use 8 for training, 2 for validation, and 2 for testing. We follow a similar experimental setting with Graber & Schwing (2020); Kofinas et al. (2021). We divide each scene into 18-step sequences. We use the first 6 time steps as input and predict the next 12 time steps. We plot MSE in fig. 4b and $L_2$ errors in fig. 21, and visualize the learned field in fig. 6, and in fig. 17 in appendix C.2.1. Since the learned field is a function of positions and orientations, we only visualize it for 4 discrete orientations, namely the group $C_4 = \left\{0, \frac{\pi}{2}, \pi, \frac{3\pi}{2}\right\}$. We visualize predictions in fig. 8b, and showcase more in appendix C.2. Again, we see that Aether outperforms all other methods. The discovered field, while hard to interpret, shows high activations that coincide with road locations *and* directions, indicating that it can guide objects through the topology of the road network.

## 5.3 GRAVITATIONAL FIELD

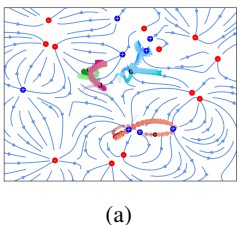 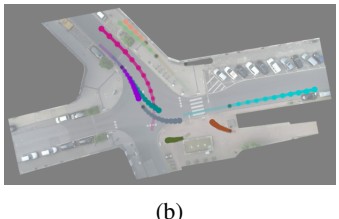 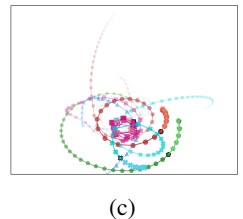

| (a) | (b) | (c) |

Figure 8: Aether predictions on (a) charged particles, (b) inD, and (c) gravity.

We now study the task of field discovery with dynamic fields, *i.e.* fields that are different across simulations. We extend the gravity dataset by Brandstetter et al. (2022) by adding gravitational sources. We create a dataset of 5,000 simulations for training, 1,000 for validation and 1,000 for testing. We use $N = 5$ particles and $M = 1$ source. We set the masses of particles to $m_p = 1$, while the source has a mass of $m_s = 10$. We generate trajectories of 49 timesteps. We use the first 44 timesteps as input and predict the remaining 5 steps.

We plot MSE fig. 4c and $L_2$ errors in fig. 22, and visualize the learned field in fig. 7. We visualize sample predictions in fig. 8c, and showcase more in appendix C.3. We observe that Aether outperforms other methods, and can accurately discover the correct "shape" of the force fields, as well as the location of the sources.

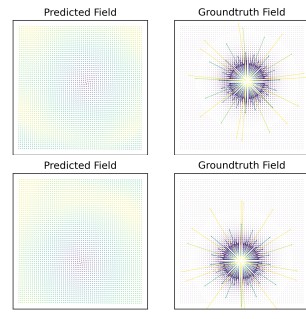

Figure 7: Learned *dynamic* fields (left) in n-body problem vs groundtruth (right).

### 5.4 ABLATION EXPERIMENTS

**Significance of discovered field** In a simulated environment like the charged particles setting, we have access to the simulator, and by extension, to the groundtruth fields and the sources that generate them. We can leverage the simulator to study the significance of the discovered field in the task of trajectory forecasting, and establish an upper bound to our performance. To that end, we create two "oracle" models that have access to the groundtruth fields, a *force oracle* and a *source oracle*. The force oracle is identical to Aether, but uses the groundtruth forces from the simulator at each point instead of predicting them with a neural field. The source oracle assumes knowledge of the "field sources". Thus, there is no longer need for disentanglement, and the problem is strictly equivariant again. We include the sources as virtual nodes in the graph, add include edges from the sources to the particles. We modify LoCS to use virtual nodes, and predict future states for all the "observable" particles, but not

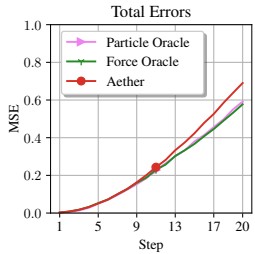

Figure 9: Results on the importance of the learned field in charged particles.

for the "source" particles. We describe this oracle in detail in appendix A.2.4. We plot the MSE in fig. 9 and the $L_2$ errors in fig. 23 in appendix D.4. We observe that our model's performance closely follows the two oracle models, and is, in fact, on par, for roughly 10 steps. This demonstrates that the discovered field is almost as helpful as the groundtruth.

## 6 CONCLUSION

In this work, we introduced *Aether*, a method that disentangles local object interactions from global field effects in interacting dynamical systems. We proposed neural fields to discover such field effects, and infer them from the dynamics alone. We combined neural fields with equivariant graph networks to learn the dynamical systems. We demonstrate that our method can accurately discover the underlying fields in a range of settings with fixed and dynamic fields, and effectively use them to forecast future trajectories.

**Limitations** Fields are not always passive, but they may react to the observable environment.

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
