# OpenReview forum: "Neural Field Discovery Disentangles Equivariance in Interacting Dynamical Systems"
_ICLR.cc/2023/Conference — Submitted to ICLR 2023_

### Official Review · Reviewer_T3HQ · 2022-10-24

**Confidence:** 3
**Correctness:** 3
**Technical Novelty And Significance:** 3
**Empirical Novelty And Significance:** 3
**Recommendation:** 5

**Clarity, Quality, Novelty And Reproducibility:**

The presentation was clear in total but I found the above unclear points. The quality was also good in total because of the above contributions and strengths. The experiments may not be reproduced at this stage (they said that their source code, data, and models will be shared upon publication).

**Details Of Ethics Concerns:**

nothing

**Strength And Weaknesses:**

Strength:
* The ideas of the above three contributions are interesting and may have a novelty.
* Overall, the paper was well-written and I can understand most of the ideas.
* The experimental results clearly show the superiority of the proposed approach.

Weakness:
* Although the presentation was overall clear, some points in methods and experimental results were unclear to me (described as the specific comments).

Specific comments
* About the name of the proposed method: Aether. The existence of Aether in modern physics may be denied, but what is the standpoint of the authors? I want to know the motivation and the opinion from the viewpoint of physics.
* In section 3.3, the author said that “Finally, in practice, we integrate G-LoCS in Aether” but in the experiments I did not find the integration. Where is this?
* First sentence in Section 5.2: InD (Bock…
* Fig 6: I understand “the learned ﬁeld is a function of positions and orientations”, but how did the author define C4 in the inD experiment? I did not understand how we should see Fig 6. The definition of the orientations here and qualitative explanations may be required.
* Section 5.4: In the ablation experiments, I did not understand why the oracle simulations accumulated errors.


**Summary Of The Paper:**

Please provide a brief summary of the paper and its contributions.
The authors proposed a disentanglement approach to local object interactions from external global ﬁeld effects, which depend on absolute positions and orientations.
The contribution of this paper is as follows:
* They introduced the idea of entangled equivariance that intertwines global and local effects, and proposed a new architecture that disentangles equivariant local object interactions from global ﬁeld effects.
* They introduced neural ﬁelds to discover global latent ﬁelds in interacting dynamical systems, and infer them by observing the dynamics alone.
* They proposed an approximately equivariant graph network that extends local coordinate frame graph networks by introducing an auxiliary origin node.
* They conducted experiments on a number of ﬁeld settings, and observed that explicitly modeling ﬁelds was mandatory for effective future forecasting, while their unsupervised discovery provided model explainability.


**Summary Of The Review:**

Based on the comment above, I consider that the strength of this paper outperformed the weakness, but I had some concerns at this stage, so I cannot give a higher rating.

---

> ### Author Response · Authors · 2022-11-15
> **Author Response to Reviewer T3HQ [1/2]**
>
> We would like to thank the reviewer for their insightful review and constructive feedback. Please find our answers below.
>
> > **About the name of the proposed method: Aether. The existence of Aether in modern physics may be denied, but what is the standpoint of the authors? I want to know the motivation and the opinion from the viewpoint of physics.**
>
> Aether was postulated since the pre-scientific era until the late 19th and early
> 20th century, when it was disproven by the Michelson-Morley experiment. We
> affectionately call our method Aether, since our method can detect "invisible"
> effects by observing dynamics alone. If aether existed, our method would be able
> to detect it. On the other hand, it could be used as a machine learning
> alternative to statistically disprove (within a statistical error margin) the
> existence of aether.
> We hope that this work will inspire the scientific community to use deep learning for field discovery.
>
> > **How did the author define C4 in the inD experiment? I did not understand how we should see Fig 6. The definition of the orientations here and qualitative explanations may be required.**
>
> Indeed, in the inD [2] experiment, the learned field is a function of positions and orientations, $\mathbf{f}: \mathbb{R}^2 \times \mathcal{S}^1 \to \mathbb{R}^2$. The orientations correspond to the object orientations (the yaw angles, as they are often referred). We follow LoCS [1] and use the angles of the velocity vectors as orientations. We use the orientations as input since we intuitively expect them to be fundamental for meaningful field prediction. Take for example, a car at the middle of a crossroad, as shown in the illustration below. Depending on whether the car is moving to the east or to the north, we would expect to have a completely different field effect that guides it along the direction of the road (or urges it to stop if there is an upcoming zebra crossing). This is in contrast to an n-body problem where the forces only depend on relative positions. Hence, learning a field that only depends on positions would result in an “averaging” across orientations and would yield poor predictions.
>
> ~~~~~
>     |     |  ↑ North
>     |     |
> -----     —---------
>         ↑
>         x→
> -----     —---------
>     |     |
>     |     |
> ~~~~~~
>
> Since we now have a continuous 3D input, it is hard to visualize the learned field on the static paper medium. Thus, we have provided a video in the supplementary material in which the input orientation is changing from 0 to $2\pi$ as time evolves. At each timestep $t \in [0, T)$, where $T$ is the duration of the video, we visualize the angle $\theta = \frac{2\pi t}{T}$. Overall, each vector shown on the grid corresponds to an $(x, y, \theta)$. The same logic applies for fig. 6 as well, except now we only visualize four input orientations that correspond to $(0, \frac{\pi}{2}, \pi, \frac{3\pi}{2})$. As an example, in the left-most subfigure, the inputs to the neural field are $(x, y, 0)$, with $x$ and $y$ being sampled on a dense grid. We have updated the figure with better descriptions and arrows that explicitly show the input orientations.
>
> > **First sentence in Section 5.2: InD (Bock...**
>
> We use a dataset named inD (Intersection Dataset) by Bock et al. [2]. We have omitted parenthesizing the citation in the submission; thank you for your comment.
> The dataset originally contains 33 recordings from real-world traffic scenes, recorded at 4 different locations.
> We created a subset that contains all scenes from a single location, namely "Frankenburg, Aachen", since this location includes the most object interactions. This allows us to study the effectiveness of static field discovery in traffic scenes.
>
> > **In section 3.3, the author said that “Finally, in practice, we integrate G-LoCS in Aether” but in the experiments I did not find the integration. Where is this?**
>
> The G-LoCS integration with Aether is detailed in Appendix A.2.1. Practically, the integration of G-LoCS in Aether means that each node state is augmented by both the origin node information, and the predicted force at the node position. We will make the description clearer in the revised version.
>
> References
> ----------
>
> [1] Miltiadis Kofinas, Naveen Shankar Nagaraja, and Efstratios Gavves. Roto-translated Local Coordinate Frames For Interacting Dynamical Systems. In Thirty-Fifth Conference on Neural Information Processing Systems (NeurIPS), 2021.
>
> [2] Julian Bock, Robert Krajewski, Tobias Moers, Steffen Runde, Lennart Vater, and Lutz Eckstein. The inD dataset: A Drone Dataset of Naturalistic Road User Trajectories at German Intersections. In 2020 IEEE Intelligent Vehicles Symposium (IV), 2020.

---

> > ### Author Response · Authors · 2022-11-15
> > **Author Response to Reviewer T3HQ [2/2]**
> >
> > > **Section 5.4: In the ablation experiments, I did not understand why the oracle simulations accumulated errors.**
> >
> > In this work, we perform ablation studies against two oracle models, a force oracle and a source oracle (incorrectly labeled as “particle oracle” in fig. 9). Both oracles have access to privileged information compared to the actual models, but they are not hardcoded to use this information correctly, or make perfect predictions. More specifically, the force oracle has access to the groundtruth forces exerted at each particle, and thus, replaces the neural field. It does *not*, however, have any information on how to actually use this information, which is passed as input to the subsequent graph network, similar to Aether. This ablation study quantitatively demonstrates that the discovered field is almost equally useful as the groundtruth one, despite the fact that they may not have identical equations that describe them.
> > We note that both oracles are trained from scratch, _i.e._ we do not merely substitute the predicted forces with the groundtruth ones in the trained model.
> >
> > On the other hand, the source oracle assumes knowledge of the source particles that generate the field. This completely removes the need for a neural field and converts the problem to a strictly equivariant one. We use a separate graph network for the source particles so that the oracle does not mix the sources with the “actual” particles. We describe this oracle in the appendix A, subsection A.2.4. Again, however, the source oracle has to learn how to use the information from other sources in this graph network. This ablation study shows that Aether can successfully predict the field and disentangle equivariance as well as an equivariant network in the strictly equivariant counterpart of the same problem.
> >
> > We hope we have addressed all of your questions and concerns, and that you consider raising your score. Please inform us if you have any remaining objections.

---

> > > ### Comment · Reviewer_T3HQ · 2022-12-05
> > > **Thank you for the response**
> > >
> > > Thank you for the response.
> > > I understood the contents of the rebuttal, but I would like to decrease the score based on the discussion with reviewers and AC (e.g., due to the simplicity for the experiments and the unclarity of how to learn the static and dynamic fields).

---

> > > > ### Author Response · Authors · 2022-12-07
> > > > **Author Response to Reviewer T3HQ**
> > > >
> > > > We would like to thank the reviewer for their response.
> > > >
> > > > > **Simplicity of experiments**
> > > >
> > > > We would like to note that the charged particles setting, whether in its original form proposed by Kipf et al. [1] or its variations, is a standard benchmark used by many state-of-the art graph networks for interacting systems and equivariant graph networks. To name a few, it has been used by Fuchs et al. (2020) [2], Satorras et al. (2021) [3], Kofinas et al. (2021) [4], Brandstetter et al. (2022) [5], and Luo et al. (2022) [6]. We follow Kipf et al., [1], and Kofinas et al. [2], and use multiple steps as input and make predictions for multiple steps as well. Our proposed dataset is at least as challenging as this dataset, and it adds _one more layer of complexity_ due to the underlying field. In fact, all the aforementioned equivariant networks will perform poorly in this new dataset.
> > > >
> > > > Similarly, the gravitational dataset was introduced by Brandstetter et al. (2022) [5]. Our variant contains fewer particles, but we make predictions for multiple steps. Once again, our proposed dataset is very challenging, since each simulation is governed by a different underlying field.
> > > >
> > > > We hope that we have addressed your concerns on the simplicity of experiments.
> > > > We consider our settings very complex, and in fact, they show that state-of-the-art equivariant methods suffer outside the idealized confines of strict equivariance.
> > > > Hence, we are curious to know what the reviewer means with the term simplicity, so that we can further improve our paper. For example, does simplicity refer to the fact that we are using two simulation datasets, or that we use relatively small graphs, or that we are using independent fields?
> > > >
> > > > References
> > > > ----------
> > > >
> > > > [1] Thomas N. Kipf, Ethan Fetaya, Kuan-Chieh Wang, Max Welling, and Richard S. Zemel. Neural Relational Inference for Interacting Systems. In ICML, 2018.
> > > >
> > > > [2] Fabian Fuchs, Daniel E. Worrall, Volker Fischer, and Max Welling. SE(3)-transformers: 3D Roto-Translation Equivariant Attention Networks. In NeurIPS, 2020.
> > > >
> > > > [3] Víctor Garcia Satorras, Emiel Hoogeboom, and Max Welling. E(n) Equivariant Graph Neural Networks. In ICML, 2021.
> > > >
> > > > [4] Miltiadis Kofinas, Naveen Shankar Nagaraja, and Efstratios Gavves. Roto-translated Local Coordinate Frames For Interacting Dynamical Systems. In NeurIPS, 2021.
> > > >
> > > > [5] Johannes Brandstetter, Rob Hesselink, Elise van der Pol, Erik Bekkers, and Max Welling. Geometric and
> > > > Physical Quantities Improve E(3) equivariant Message Passing. In ICLR, 2022.
> > > >
> > > > [6] Shitong Luo, Jiahan Li, Jiaqi Guan, Yufeng Su, Chaoran Cheng, Jian Peng, and Jianzhu Ma. Equivariant Point Cloud Analysis via Learning Orientations for Message Passing. In CVPR, 2022.

---

### Official Review · Reviewer_Gz6i · 2022-10-26

**Confidence:** 3
**Correctness:** 3
**Technical Novelty And Significance:** 3
**Empirical Novelty And Significance:** 3
**Recommendation:** 6

**Clarity, Quality, Novelty And Reproducibility:**

Clarity: This paper is well-written overall. I do have a question about section 3.3. How is it related to section 3.1?

Novelty: I think this is a nice paper as it is studying the force field discovery in a more practical setting.

**Strength And Weaknesses:**

Strength:

While previous works focus on the discovery of the interacting force between particles from the observable trajectory of particles, this work takes a step further by considering the existence of an underlying global force field. This generalized setting cannot be captured by the equivariant networks as they fail to capture global information. By incorporating an additional neural field, the authors design a new model, Aether, which in principle is able to automatically learn the interacting force and global force separately, which hence disentangles these two forces.

Weakness:

1. The global field is assumed to be independent of the observed objectives, but this seems to be a strong assumption, e.g. when the blackhole is considered as the source of the global field, its position and velocity are also affected by the planets around it. Therefore, this assumption may not hold, especially in a large time interval.
2. While authors do consider the setting of dynamic field, it is clear how the interacting force can be disentangled from this time-varying dynamic field. In the worst case scenario, all forces can be described by a dynamic field.

**Summary Of The Paper:**

This paper considers the task of discovering the underlying force field of the interacting dynamical systems. In particular, this paper focus on the setting where the observed trajectories of particles are driven by the interacting forces between the particles as well as a global force field. The entanglement of global and local effects makes recently popularized equivariant networks inapplicable, since they fail to capture global information. To address this issue, this work proposes to disentangle local object interactions from external global field effects. Specifically, a neural field is used to approximate the latent field and the interactions between particles are modeled with equivariant graph networks operating in local coordinate frames. Experiments show that the proposed approach accurately discovers the underlying fields in several interesting application.

**Summary Of The Review:**

This is nice paper that studies the force field discovery in a more practical setting. A question I have for the authors is that in the more interesting dynamic field setting, why can we expect the dynamic force and the interacting forces to be disentangled using this approach? Intuitively, this is not possible without further assumptions.

---

> ### Author Response · Authors · 2022-11-15
> **Author Response to Reviewer Gz6i**
>
> We would like to thank the reviewer for their insightful review and constructive feedback. Please find our answers below.
>
> > **The global field is assumed to be independent of the observed objectives, but this seems to be a strong assumption, e.g. when the blackhole is considered as the source of the global field, its position and velocity are also affected by the planets around it. Therefore, this assumption may not hold, especially in a large time interval.**
>
> A lot of fields in science are passive. Examples close to our work include static electric or magnetic fields, while the gravitational field of the earth is also a static field. Another example that falls outside the scope of this work includes the --independent-- boundary conditions in an interacting dynamical system.
>
> Apart from fields that are actually static, many others can be effectively modelled as such.
> It is true, as the reviewer suggests, that this assumption may not hold for larger time intervals, but it is often a good modelling choice for relatively short time spans.
> A well-known example from physics includes the sun-earth-moon system.
> If we consider the gravitational field created by the Sun in our solar system, it is not passive, rather it depends on the positions of the other celestial objects. In practice, however, the effect of all these objects can be negligible, such that simulations that ignore them can achieve very high accuracy.
>
> As mentioned [in our common responses](https://openreview.net/forum?id=wZRgC1McxyU&noteId=QUHYrnpfSBL), discovering non-independent fields is definitely our main inspiration and ultimate goal. We consider it beyond the scope of this work because we aim at addressing an intermediate very challenging --when described within a learning framework and without assuming specialized knowledge-- problem: unsupervised field discovery in interacting systems.
> We note that our work is, to the best of our knowledge, the only work that successfully performs the task of unsupervised field discovery in interacting systems, a novelty that was perhaps not stressed out enough in the paper. We hope that this work will inspire the community and bootstrap a line of works that focuses on more and more realistic field discovery, _since fields are omnipresent in all scientific tasks_.
>
> > **How is section 3.3 related to section 3.1?**
>
> In section 3.3 we introduce G-LoCS, a simple approximately equivariant graph network that combines local and global states. This network incorporates an auxiliary graph node that corresponds to the global coordinate frame. This simple modification enables the network to make use of global information, should it be relevant to the task at hand. We note, however, that G-LoCS cannot disentangle equivariance, since that would require an explicit modelling of global field effects.
> This section serves two purposes. First, it introduces an alternative to Aether that is capable of exploiting both local interactions and global information. Second, G-LoCS can be seen as complementary to Aether; it can replace the equivariant graph network that follows field discovery and performs trajectory forecasting.
>
> > **In the more interesting dynamic field setting, why can we expect the dynamic force and the interacting forces to be disentangled using this approach? Intuitively, this is not possible without further assumptions.** [...] **While authors do consider the setting of dynamic field, it is clear how the interacting force can be disentangled from this time-varying dynamic field. In the worst case scenario, all forces can be described by a dynamic field.**
>
> While the dynamic field setting is far more challenging than the static field setting, it is certainly possible to disentangle local interactions from global field effects. Taking the gravitational field as an example, even thought the field sources are different in each simulation, the dynamics are still invariant, that is governed by the same underlying law, namely an inverse square law. Furthermore, the forces between observable particle pairs are always co-linear to their relative positions. This makes disentanglement feasible, since the field force can be computed as the vector subtraction between the observable net effect and the equivariant effect.
> Of course, the example above is a simplification, since the network has to figure these physical rules out, all of which are non-trivial. Nevertheless, this argument intuitively explains how _disentanglement is possible_ in a dynamic field setting.
>
> We hope we have addressed all of your questions and concerns, and that you consider raising your score. Please inform us if you have any remaining objections.

---

### Official Review · Reviewer_DdmJ · 2022-10-29

**Confidence:** 3
**Correctness:** 3
**Technical Novelty And Significance:** 3
**Empirical Novelty And Significance:** 3
**Recommendation:** 5

**Clarity, Quality, Novelty And Reproducibility:**

clarity is high, except the core method
quality is good, easy to understand
novelty is not clear, may need improvement
reproducibility, is not sure. The domain knowledge of dynamic systems is needed and no code is provided.


**Details Of Ethics Concerns:**

No concerns

**Strength And Weaknesses:**

- Strength

1. The paper is well-written and organized so that it is easy to follow.
2. Background section provides many necessary details and information
3. The overall idea of disentangling local and global is very interesting

- Weakness

1. The novelty is not clear to show.
2. Lack of high-level intuition why the proposed method works better, specifically disentanglement property


**Summary Of The Paper:**

This work introduces Aether, a new method that disentangles local object interactions from global field effects in interacting dynamic systems. The new approach leverages the neural field and VAE model to infer from the dynamics along. Then combining neural fields with equivariant graph networks learns dynamic systems.  A set of examples are used to demonstrate the methods compared with several baseline methods.

**Summary Of The Review:**

Overall, the paper is good to show the idea with solid experiments. However, the core idea is not very clear to show. As I mentioned in the weakness,

1. what's the core difference between aether and previous SOTA work？
2. what's the major contribution that makes entanglement to disentanglement?
3. What's high-level intuition?
4. The method is VAE-based disentanglement. Any previous related works?
5. What's the current limitation, except the domain limitation?

---

> ### Author Response · Authors · 2022-11-15
> **Author Response to Reviewer DdmJ [1/2]**
>
> We would like to thank the reviewer for their insightful review and constructive feedback. Please find our answers below.
>
> > **Unclear novelty**
>
> See [our response to common questions](https://openreview.net/forum?id=wZRgC1McxyU&noteId=QUHYrnpfSBL).
>
> > **What is the core difference between Aether and previous SotA works?**
>
> All strictly equivariant graph networks are inapplicable in settings that involve global effects. This is not a practical limitation, rather it is true by design; equivariant neural networks can only capture local states. Our method, Aether, _is the first work_ that can model interacting systems with global effects. Furthermore, _we are the first_ to discover the underlying fields in interacting dynamical systems.
>
> > **The method is VAE-based disentanglement. Any previous related works?**
>
> While our notions of entangled equivariance and disentangling equivariance bear similarities with disentanglement in the context of VAEs and generative models, the two are quite distinct. First, in this work, we have introduced the notion of "entangled equivariance". This refers to the fact that while a dynamical system might be governed by equivariant dynamics (e.g. forces between pairs of objects), it can also be affected by global, non-equivariant effects. What we observe is the net force from these two constituents; we name this concept _entangled equivariance_. Second, we propose a neural field that discovers the global effects, and a graph network that focuses on equivariance. Thus, we say that we can _disentangle equivariance_.
> The similarity with VAE disentanglement is the fact that we \emph{explicitly} include the predicted global forces in the state description of each node-object pair, alongside the equivariant features.
>
> While our method can be combined with any equivariant graph network, in practice, our full model contains a VAE-based equivariant graph network that follows the NRI [1], dNRI [2], LoCS [3]. This VAE comprises an encoder that predicts latent interactions between object pairs, and a decoder that uses the latent graph structure to make predictions. Our VAE, however, _does not do any disentanglement_.
>
> Overall, while there are a lot of works in the literature that research the topic of disentanglement in VAEs, our work introduces the notion of entangled equivariance, which is very common in dynamical systems, since global fields are omnipresent in many scientific tasks. While geometric deep learning has been evolving rapidly in the last few years, state-of-the-art strictly equivariant networks are inapplicable in settings with entangled equivariance. *We are the first* to propose a method that can model such systems, by learning the underlying global field, which also allows us to disentangle equivariance.
>
> We understand that the term disentanglement can be misleading for the reader. We will explicitly mention the differences with VAE disentanglement in the revised version.
>
> > **What is the current limitation, except the domain limitation?**
>
> Our method is limited in the type of fields in can discover. Namely, we have only considered passive fields, _i.e._ fields that do not react to the observable environment. Many fields in science are passive, while many others can be effectively modelled as such. For example, if we consider the gravitational field created by the Sun in our solar system, it is not passive, rather it depends on the positions of all other celestial objects. In practice, however, the effect of all these objects can be negligible, such that simulations that ignore them can achieve very high accuracy. In other scenarios, however, active fields might be crucial for effective field discovery and modelling of the system dynamics. We aspire to explore active fields in future work.
>
> Another limitation is that, in the dynamic field setting, we assume that the input trajectories for each scene are perfectly descriptive for the field we are trying to discover, and we do not need multiple scenes evolving in the same field to uncover it. While this hypothesis holds for a large number of settings, it might not always be true. Future work can explore this very interesting research direction.
>
> References
> ----------
>
> [1] Thomas N. Kipf, Ethan Fetaya, Kuan-Chieh Wang, Max Welling, and Richard S. Zemel. Neural Relational Inference for Interacting Systems. In ICML, 2018
>
> [2] Colin Graber and Alexander G. Schwing. Dynamic Neural Relational Inference. In CVPR, 2020.
>
> [3] Miltiadis Kofinas, Naveen Shankar Nagaraja, and Efstratios Gavves. Roto-translated Local Coordinate Frames For Interacting Dynamical Systems. In NeurIPS, 2021.

---

> > ### Author Response · Authors · 2022-11-15
> > **Author Response to Reviewer DdmJ [2/2]**
> >
> > > **Domain knowledge of dynamical systems in needed.**
> >
> > While a number of works on interacting dynamical systems delves deep in the rich literature of dynamical systems, our work follows a more deep learning-oriented approach to modelling such systems and forecasting future trajectories. As such, the background section that introduces interacting systems and their geometric graph formalism, as well as the description of the learning problem, namely future foreacasting, should lay out all necessary knowledge to understand our work.
> >
> > > **Reproducibility might need improvement.**
> >
> > We have updated the supplementary material (see Appendix A.2 in the revised version) to include all necessary details to reproduce our work.
> >
> > We hope we have addressed all of your questions and concerns, and that you consider raising your score. Please inform us if you have any remaining objections.

---

### Official Review · Reviewer_HEK4 · 2022-11-02

**Confidence:** 3
**Clarity, Quality, Novelty And Reproducibility:** See strength and weakness section.
**Correctness:** 3
**Technical Novelty And Significance:** 3
**Empirical Novelty And Significance:** 3
**Recommendation:** 6

**Strength And Weaknesses:**

Strengths:

1. This work proposes a novel notion of disentangled equivariance, and proposes a framework to successfully disentangle global field effects from local system interactions that are equivariant to translations and rotations.
2. They are able to predict global field effects that are latent by observing the system dynamics using neural fields.

Weaknesses:

1. The paper need more work to explain the motivation and their method. Most of the explanation for base concepts is referred to in related papers.
2. The experimental setup is not clear. Again, an explanation of datasets and learning problem should have been provided, either in the main paper or if not, in the supplementary.
3. The relationship between the notion of approximate equivariance and disentangled equivariance is unclear from their description. In section 3.3, they mention that the difference lies in the fact that approximate equivariance tries to combine global field and local field effects. But it seems from their model description in section 3.1 that they are doing the same thing (especially from eqs. 5-7). Hence, the novelty of their notion of disentangled equivariance is unclear.
4. The experiments are also conducted on fairly limited settings (in two dimensional systems) and in scenarios where the underlying global fields remain fixed throughout the interaction. This makes it difficult to assess the impact of this work.
5. The results are also difficult to interpret and require more details about what is the ideal result and how the method compares to the baselines. For eg., in Figure 7, it is unclear what the figures represent and why the predicted and ground-truth fields look so different.

Other things that could improve the paper:

1. A more detailed description of their experimental setup, and datasets used in the supplementary would be helpful to making the method and its impact more clear.
2. Deeper explanation of the model architecture by using the supplementary material would also be helpful.

**Summary Of The Paper:**

This paper proposes the notion of disentangled equivariance for systems of interactive systems, as well as a framework that disentangles the effects of underlying global field from the local system dynamics that are equivariant to transformations in SE(3). They are also able to learn a neural field to predict underlying field effects.

**Summary Of The Review:**

See strength and weakness section.
------------
The authors' response address many of my concerns and questions. I will raise the score.

---

> ### Author Response · Authors · 2022-11-15
> **Author Response to Reviewer HEK4 [1/2]**
>
> We would like to thank the reviewer for their insightful review and constructive feedback. Please find our answers below.
>
> > **Motivation and novelty**
>
> See [our response in common questions](https://openreview.net/forum?id=wZRgC1McxyU&noteId=QUHYrnpfSBL).
>
> > **The experiments are conducted on fairly limited settings (in two dimensional systems) and in scenarios where the underlying global fields remain fixed throughout the interaction.**
>
> Discovering non-fixed fields is definitely our main inspiration and ultimate goal. We consider it beyond the scope of this work because we aim at addressing a very challenging --when described within a learning framework and without assuming specialized knowledge-- intermediate problem: unsupervised field discovery in interacting dynamical systems.
>
> On the other hand, 3D fixed fields can be modelled by our method in a very straightforward way, by just changing the number of input and output variables. To demonstrate this, we extend the gravitational field dataset used in our work in 3 dimensions. The dataset consists of 5,000 simulations for training, 1,000 for validation and 1,000 for testing. We use N = 5 particles and M = 1 source. Each simulation lasts for 49 timesteps; we use the first 44 timesteps as input and predict the remaining 5 steps. In the table below we report the MSE@5, i.e. the MSE at the last prediction timestep; we will add the full results in the revised version. Our method clearly outperforms all baselines.
>
> | Method |Aether |G-LoCS |LoCS  |dNRI  |
> | :----- |:----- |:----- |:---- |:---- |
> | MSE@5  |__0.339__  |0.439  |0.464 |0.568 |
>
> > **The relationship between the notion of approximate equivariance and disentangled equivariance is unclear from their description. In section 3.3, they mention that the difference lies in the fact that approximate equivariance tries to combine global field and local field effects. But it seems from their model description in section 3.1 that they are doing the same thing (especially from eqs. 5-7). Hence, the novelty of their notion of disentangled equivariance is unclear.**
>
> Many settings that involve interacting systems, such as the n-body problems we study, would be strictly equivariant if we had access to all sources of information, including the “external sources” that generate the field. In practice, though, we can only observe a part of the whole system that does not include these “external forces”.
>
> Wang et al. [1] use the definition of equivariance error to define approximate equivariance. Equivariance error is a commonly used metric in equivariant deep learning, previously used to measure errors due to discretization, low order approximation, or interpolation artifacts.
>
> By contrast, we consider entangled equivariance as a property of the setting/problem/data. For example, in n-body problems in a (hidden) gravitational field, the interactions between objects follow certain symmetries, and are strictly equivariant, but the net forces we observe contain non-equivariant constituents.
> While approximate equivariance could also be used to describe
> such settings, the two concepts are distinct and this would overload nomenclature.
>
> Importantly, we can clearly demonstrate the difference between _disentangling equivariance_ and _approximate equivariance_, and show that are not strict subsets of one another, by answering the question "Can all approximately equivariant networks disentangle equivariance?". *The answer here is **no***. Disentangling equivariance means that we explicitly separate and explicitly describe the non-equivariant forces or effects caused by global states.
>
> In our work, we have presented both an approximately equivariant network, and an equivariance disentangling network.
> In section 3.3, we propose G-LoCS, an *approximately equivariant network*, that directly combines global and local information.
> In and of itself, however, G-LoCS is not able to disentangle equivariance.
> Our neural field, paired with an equivariant (or approximately equivariant) network can disentangle equivariance,
> and the whole model, Aether, outperforms G-LoCS in all settings.
> More specifically, the very fact that we explicitly predict and use the forces in equations 5-7, is what makes us able to disentangle equivariance.
> Our neural field predicts the dynamics constituents that break strict equivariance, and moves us back to the equivariant regime. In contrast, equations 11-12 use the global states of nodes, but do not explicitly model these non-equivariant constituents.
>
> We note that we have erroneously named section 3.3 "Disentangling equivariance with global-local coordinate frames", which we have changed to "Approximate equivariance with global-local coordinate frames" in the revision.
>
> References
> ----------
>
> [1] Rui Wang, Robin Walters, and Rose Yu. Approximately Equivariant Networks for Imperfectly Symmetric
> Dynamics. In ICML, 2022.

---

> > ### Author Response · Authors · 2022-11-15
> > **Author Response to Reviewer HEK4 [2/2]**
> >
> > > **An explanation of the learning problem should have been provided.**
> >
> > We are interested in learning interacting dynamical systems, and more specifically in the task of trajectory forecasting. Our inputs comprise a set of $N$ object states for $T$ timesteps, $\mathbf{X} \in \mathbb{R}^{N \times T \times D}$. The state $\mathbf{x}_i^t$ describes the position and the velocity of the $i$-th object at timestep $t$. Given these inputs, our task is to predict the future states (positions and velocities) for all objects and for a number of timesteps. See figure 3 also for an illustration of the inputs and outputs to our problem. We will update the manuscript in the revision to explain the learning problem more explicitly.
> >
> > > **The results are difficult to interpret and require more details about the ideal result and how the method compares to baselines.**
> >
> > We have updated the qualitative results for all datasets in the supplementary material, appendix C, along with more details on how to interpret them. We have laid out our predictions next to predictions from baselines for a direct comparison. In the charged particles setting and in the gravitational setting, we visualize the groundtruth on each plot, while on inD we visualize the groundtruth in a different column. In all settings, it is clear that our method makes better predictions that respect the underlying fields.
> >
> > > **In Figure 7, it is unclear what the figures represent and why the predicted and ground-truth fields look so different.**
> >
> > Our model performs unsupervised field discovery, i.e. discovers the underlying fields in interacting systems of objects without ever directly observing the groundtruth field --*not even as a supervisory signal*. Furthermore, the discovered field is used as input to a graph neural network. This means that the discovered field can greatly vary from the groundtruth and still be useful for the network.
> >
> > We motivate this with the following example: a force field with inverse magnitude $\frac{1}{\|\mathbf{f}\|}$ would be an "equally correct" prediction, since the subsequent neural network can reverse the predictions.
> > In fig. 7, the learned field indeed seems to be close to an inverse magnitude force field
> > $\frac{1}{\|\mathbf{f}\|}$. We hypothesize that this can be an *even better* representation for the graph network that follows, because the inverse magnitude is more learning-friendly; it approaches zero close to the field sources, areas where precise predictions are most important. Since floating-point precision is best close to zero, this can be an advantage for the
> > neural network overall.
> >
> > > **A more detailed description of their experimental setup, and datasets used in the supplementary would be helpful to making the method and its impact more clear.**
> >
> > We have updated the supplementary material (see Appendix B) to better describe our experimental setup and datasets used in this work.
> >
> > > **Deeper explanation of the model architecture by using the supplementary material would be helpful.**
> >
> > We have updated the supplementary material (see Appendix A.2 in the revised version) to explain our model architecture in greater depth.
> >
> > We hope we have addressed all of your questions and concerns, and that you consider raising your score. Please inform us if you have any remaining objections.

---

### Author Response · Authors · 2022-11-15
**Author response to common questions**

We would like to thank all reviewers for their insightful reviews and constructive feedback. Below we answer some commonly asked questions.

> **Motivation**

Fields are omnipresent in the real world, they appear, for instance, in electromagnetism, astronomy and cosmology, fluid dynamics, as well as in social dynamics. While modelling interacting dynamical systems is downright complex on its own right,
leading to several relevant state-of-the-art deep learning-based approaches in recent years,
the effect of existing yet unobservable fields has not been studied to date.
For instance, how can we model charged electric particles under the influence of an underlying electrostatic field,
or how can we model social dynamics under the influence of a traffic flow field?

Hence, the motivation of this paper is modelling complex interacting dynamical systems evolving under the influence of latent fields, learning the inverse problem of recovering those fields, and learning to use those fields to better forecast future trajectories.

In the process of modelling latent fields, we arrive at our second motivation of disentangling equivariance: assuming void ambient spaces and perfect conditions allows for modelling equivariance. In the presence of latent fields, however, equivariance is not directly applicable, and can only be reinstated after disentangling the local object interactions from the underlying field effects.

> **Novelty is unclear**

Following our motivation, our novelties are:

1. We introduce neural fields to discover global latent fields in interacting dynamical systems, and infer them by observing the dynamics alone.
2. We introduce the notion of entangled equivariance that intertwines global and local effects.
3. We propose a novel architecture that disentangles equivariant local object interactions from global field effects.
3. We propose a simple approximately equivariant graph network as an alternative, or complementary to equivariance disentanglement.

We note that our work is, to the best of our knowledge, the only work that successfully performs the task of unsupervised field discovery in interacting systems, a novelty that was perhaps not stressed out enough in the paper. We hope that this work will inspire the community and bootstrap a line of works that focuses on more and more realistic field discovery, _since fields are omnipresent in all scientific tasks_.

---

### Author Response · Authors · 2022-11-18
**Kind reminder for reviewers**

Dear reviewers,

We are entering the last day of the discussion period. 3 days ago, we responded to each reviewer's points and uploaded a revised version of our paper and our supplementary material that contains multiple improvements motivated by the reviewers' points.

We would be thankful if the reviewers could take a look at our responses and update their reviews if we have cleared up the remaining concerns. If not, we would be happy to incorporate further changes in the camera-ready version of our paper!

---

### Decision · Program_Chairs · 2023-01-20

**Decision:**

Reject

**Justification For Why Not Higher Score:**

This was a very clear case.

**Justification For Why Not Lower Score:**

-

**Metareview: Summary, Strengths And Weaknesses:**

This paper has received four expert reviews and was a borderline case. The paper was discussed first asynchronously over openreview and finally a meeting was held between the AC and two of the reviewers, which had initially given borderline ratings. The discussion of the paper again brought several issues to light:

- The experiments and tasks are extremely simple, and far from the SOTA.
- Generalization experiments are missing, for instance increasing the number of objects / particles between training and testing. This is important as this is one part which is captured by an equivariant model with pairwise terms in a GNN only, and the additional global field could hurt this capacity.
- It is unclear how estimating the field is done ensuring that disentangling is done during training. The problem is that a part of the model (the field) seems to have been learned from data which corresponds to the full problem, including the equivariant part.
- There is no comparison with the SOTA on models which could solve this problem, in particular what the authors call "Approximate equivariance". The papers contains a full related work paragraph on them, but it is not clear why the proposed method cannot be compared to this body of work.

After the meeting the AC interacted with the authors regarding learning the non-equivariant (global) field from the full data only. This discussion confirmed that the authors were (1) not aware of this quite important technical problem and, had not addressed it. This confirms a major weakness of the paper, as learning the global (non-equivariant) field alone from full data (even if it considered as "given") is an option, and the signal processing and ML community has been dealing with similar issues for a long time, for instance through alternating optimization schemes.

A consensus had already emerged before, and together with the other shortcomings of the paper, this lead to a clear decision of rejection.

**Summary Of Ac-Reviewer Meeting:**

Explained in the meta-review